# Attention-Based Deep Learning System for Classification of Breast Lesions—Multimodal, Weakly Supervised Approach

**DOI:** 10.3390/cancers15102704

**Published:** 2023-05-10

**Authors:** Maciej Bobowicz, Marlena Rygusik, Jakub Buler, Rafał Buler, Maria Ferlin, Arkadiusz Kwasigroch, Edyta Szurowska, Michał Grochowski

**Affiliations:** 12nd Department of Radiology, Medical University of Gdansk, 80-214 Gdansk, Poland; marlena.rygusik@gumed.edu.pl (M.R.); edyta.szurowska@gumed.edu.pl (E.S.); 2Department of Intelligent Control Systems and Decision Support, Faculty of Electrical and Control Engineering, Gdansk University of Technology, 80-233 Gdansk, Poland; s176315@student.pg.edu.pl (J.B.); s176352@student.pg.edu.pl (R.B.); maria.ferlin@pg.edu.pl (M.F.); arkadiusz.kwasigroch@pg.edu.pl (A.K.)

**Keywords:** breast cancer, digital mammography, artificial intelligence, machine learning, weakly supervised learning, multimodal learning, attention maps, scarcity of data, breast lesion classification

## Abstract

**Simple Summary:**

Breast cancer affects millions of women worldwide. We aim to provide radiologists with automatic support for mammography review. We propose deploying deep learning models with Multiple Instance Learning algorithms for breast cancer diagnosis (cancer versus non-cancer classification) based on digital mammography images, taking advantage of data annotated at an image level without indicating where the lesion is. We employed algorithms to analyse original, high-resolution images with minimal reduction in size. We used graphical suggestions of attentional maps to verify the correctness of the algorithm and to indicate the areas of the images where cancer lesions are. Finally, we performed comparative and validation studies on external datasets differing in the number of images, pixel intensity levels, and subtypes of existing lesions, showing high accuracy and potential for generalisability of the algorithms.

**Abstract:**

Breast cancer is the most frequent female cancer, with a considerable disease burden and high mortality. Early diagnosis with screening mammography might be facilitated by automated systems supported by deep learning artificial intelligence. We propose a model based on a weakly supervised Clustering-constrained Attention Multiple Instance Learning (CLAM) classifier able to train under data scarcity effectively. We used a private dataset with 1174 non-cancer and 794 cancer images labelled at the image level with pathological ground truth confirmation. We used feature extractors (ResNet-18, ResNet-34, ResNet-50 and EfficientNet-B0) pre-trained on ImageNet. The best results were achieved with multimodal-view classification using both CC and MLO images simultaneously, resized by half, with a patch size of 224 px and an overlap of 0.25. It resulted in AUC-ROC = 0.896 ± 0.017, F1-score 81.8 ± 3.2, accuracy 81.6 ± 3.2, precision 82.4 ± 3.3, and recall 81.6 ± 3.2. Evaluation with the Chinese Mammography Database, with 5-fold cross-validation, patient-wise breakdowns, and transfer learning, resulted in AUC-ROC 0.848 ± 0.015, F1-score 78.6 ± 2.0, accuracy 78.4 ± 1.9, precision 78.8 ± 2.0, and recall 78.4 ± 1.9. The CLAM algorithm’s attentional maps indicate the features most relevant to the algorithm in the images. Our approach was more effective than in many other studies, allowing for some explainability and identifying erroneous predictions based on the wrong premises.

## 1. Introduction

Breast cancer, with more than 2.2 million new cases in 2020, is the most common cancer in women worldwide [1]. It is responsible for around 680,000 cancer-related deaths, despite the rapid improvement in diagnostics and treatment methods [1]. Currently, the diagnosis is based on clinical examination, mammography, ultrasonography of the breast and loco-regional lymph nodes, and in some cases, the magnetic resonance of the breast. These examinations are followed by invasive breast tumour and lymph node biopsy procedures and pathological examination of specimens to verify the clinical diagnosis. There is no faster, less invasive, and cheaper diagnostic approach so far, which could reduce the number of necessary radiological, surgical, and pathology tests to confirm the presence of cancer and its potential metastases [2]. The most crucial task is to detect any suspicious lesion in the first examination, such as screening mammography, and then decide whether it might be malignant [3,4].

Mammography screening, widely used for the early detection of breast cancer, generates millions of film or digital mammograms requiring vast numbers of very well-trained radiologists to assess them. For many years, several attempts were made to create computer-aided diagnosis (CAD) support systems reading mammograms to facilitate more automated detection of breast lesions and their classification [5,6,7]. These systems were based on machine learning algorithms, mostly on deep learning. Our recent search with IEEE Xplore with terms ‘breast cancer’, AND ‘mammography’ AND ‘machine learning’, OR ‘deep learning’, OR ‘artificial intelligence’ OR ‘computer aided diagnosis’, OR ‘algorithms’ returned over 1500 records from years 2016 to 2022 with over 250 papers and conference proceedings addressing artificial intelligence in breast cancer diagnosis and classification based on film or digital mammography.

Existing deep learning algorithms can reach human-level or almost human-level performance in many tasks [8,9,10,11,12]. Despite the great success of AI-based computer-aided diagnosis systems, there are many problems related to using these algorithms. One of the significant difficulties is the insufficient number of adequately segmented and labelled data to train such large-scale models effectively. Another difficulty is selecting an appropriate structure for neural models and accompanying mechanisms supporting data analysis. Finally, and most importantly, clinicians have to deal with the complexity and frequently unclear interpretations of decisions taken by those data-driven, black-box models. The latter is essential considering various types of biases and artefacts in the datasets [13]. Until these issues are addressed, the broad adoption of deep learning-based decision support systems in clinical practice is limited.

In this paper, we would like to familiarise the readers with our research towards AI-based computer-aided diagnosis system for the early detection and diagnosis of breast cancer. To ensure the trustworthiness of the system’s performance, we would like the system to indicate the fundamental premises for a given decision. The assumption is that the data are annotated only at the image level, so it is impossible to take advantage of systems for segmentation. Furthermore, the data available for analysis are relatively limited. For this purpose, we designed a weakly supervised Multiple Instance Learning deep learning system exploring the information hidden in the mediolateral oblique (MLO) and craniocaudal (CC) full-field digital mammography images. Such an approach makes it possible to train the system under data scarcity effectively. It requires limited involvement of highly skilled breast radiologists in the annotation process but involves them in the explainability and information feedback loops. The designed system has been comprehensively validated for using different types and structures of analysed data, the impact of hyperparameters, and the detection and impact of artefacts. Our research used the MUG dataset from the Medical University of Gdansk, Poland, and the Chinese Mammography Database (CMMD). All the analyses were conducted on the MUG dataset, and finally, the optimised structure of the CAD system was validated on the CMMD dataset.

The contributions of the paper include the following:A proposition for the deployment of deep learning models with Multiple Instance Learning algorithms for breast cancer diagnosis based on full-field digital mammography MLO and CC images, taking advantage of data annotated just at an image level;A comparative analysis of breast cancer diagnosis performance based on just CC images, just MLO images, the analysis of combined CC and MLO images, and multimodal analysis of CC and MLO images;An analysis of the potential of employing attentional maps to verify the correctness of the algorithm and to indicate the areas of the images where cancer foci are located (detection);Comprehensive ablation studies showing the impact of the system’s hyperparameters on the effectiveness of the system’s performance, quality, and usability for the end users of the accompanying attentional maps. Most common neural models were also investigated;Analysis and discussion of the results confirming the classification effectiveness of the proposed approach and showing the benefits of using attentional maps to detect biases in the data causing seemingly effective but completely wrong system operation;Comparative and validation studies on sets differing in the number of images, pixel intensity levels, and subtypes of existing lesions;An analysis of the benefits of transfer learning from one dataset to another.

## 2. Related Work

In recent years, we have witnessed substantial research on systems to automatically support the diagnosis and analysis of breast cancer lesions—computer-aided diagnosis systems. These include the classification, detection, and segmentation of breast tumours [14,15,16,17]. Such systems mainly exploit algorithms from the broad family of artificial intelligence, particularly deep learning algorithms.

Often, their effectiveness is on par with that of experienced radiologists, and almost always, such systems contribute to the effectiveness of clinicians if designed and used appropriately. According to [14], more than 70% of research work for breast cancer analysis is conducted on mammograms. The standard mammography examination consists of two projections for each breast—CC and MLO—and most of the research work considers these views separately.

In [18], Petrini et al., proposed a two-view classifier as a pair of single-view classifiers with the upper layers removed, followed by a concatenation of the outputs for both views. The single-view classifier was created by learning a patch classifier and modifying the top layers. The patch classifier was learned on ten patches selected from a given region of interest (ROI) and another ten from the background with a patch size of 224 × 224 px for each breast mammogram. Afterwards, the single-view model was trained on full-size mammograms (1152 × 896 px). The dual-view approach has improved performance, increasing AUC from 0.8033 ± 0.0183 to 0.8418 ± 0.0258.

Feature fusion of two-view mammograms was considered in [19] by Li et al. The authors performed a binary classification task (benign, malignant) using one patch cut from the annotation area. The size of the patch originally depended on the dimensions of the cut breast mass and was eventually changed to 512 × 512 px. The authors proposed two separate CNN models that extract CC and MLO patch features pre-trained with MLO and CC patches, respectively, and then fine-tuned with CC and MLO patches, respectively. All convolutional, pooling, and fully connected layers parameters were then frozen. Finally, two gate recurrent unit (GRU) modules that shared the same parameters in the fine-tuning process fused features from two-view mammograms. The fusion of the two views’ features improved accuracy from 91.0% and 90.8% for CC and MLO, respectively, considered separately, to 94.7% when fused.

Deep neural networks (DNN) are naturally predisposed to efficiently handle vast amounts of data, including medical data. However, many issues related to the effective and, importantly, trustworthy use of AI technologies must be tackled. The most common and effective DNNs are very large. For instance, the popular DNN architecture ResNet-50 consists of 48 convolutional layers along with 1 MaxPooling and 1 AveragePooling layer, which brings a total of 25.6 million parameters [20]. Another example is EfficientNet-B0, which is mainly built with mobile inverted bottleneck blocks, convolutional layers, one pooling layer, and a fully connected layer with about 5.3 million parameters as a whole [21]. We often take advantage of supervised learning to optimise the parameters of the neural models. It requires a considerable amount of good-quality and balanced training data.

However, in the case of many domains, including medicine, collecting data, especially annotated, is extremely costly, time-consuming, and thus, tricky. The main reasons are legal and privacy restrictions [22,23,24] and the need for manual annotations of the datasets, requiring much time from high-level medical experts.

In particular, the main challenge is not the availability of data but the availability of their annotations at the required precision level [25]. Manual data labelling by medical experts is laborious, time-consuming, and subjective [26,27]. Hence, in medical practice, we often experience situations where only a tiny part of data is precisely marked and labelled at the pixel level. At the same time, the rest is described just at the image level [26,28,29]. Different regularisation methods, such as data augmentation [30,31,32], or approaches to model training, such as unsupervised, weakly, or self-supervised learning, are used to cope with such conditions [33,34,35,36]. Given the nature of medical diagnostics, mainly the requirement for high performance and reliability, a supervised learning approach is still the best choice. Nevertheless, it is vital to seek strategies that minimise mentioned drawbacks and risks.

Another significant problem is that deep learning structures are black box models that map a given input to a target output [37,38,39]. Due to the complexity of the architecture, the large number of parameters and the specificity of the algorithms employed, they suffer from a lack of transparency during the training and decision-making processes. This lack of transparency during the decision-making process and unclear rules of reasoning concern end-users and make it difficult or sometimes even impossible to safely use it in many fields of our lives, especially in medicine.

The general lack of trust is additionally intensified by several cases in which DNNs made decisions based on wrong deduction or because it was fooled by an adversarial attack [40]. Even the EU General Data Protection Regulation states that every person has the right to an explanation, which means that DNN-based systems should provide explanations of decisions being undertaken [23,41].

The other important issue is that DNN training may be affected by irrelevant noisy areas or unwanted artefacts in the images [13]. Detailed analysis of the reasoning process can help to find and eliminate errors in deduction, to prove that DNN is working as expected, or even to extract new and essential features unnoticed by humans [41,42,43].

Most of the work on deep learning interpretability has focused on understanding the grounds of decisions [38,41,44]. In computer vision, interpretability is achieved mainly by visualisation techniques, verbal explanations, and analytical explanations [44,45]. The subcategory of visual explanations covers methods such as attention maps, visual saliency in heatmaps, and visualising class-related patterns [38,46]. An exciting branch of visual explanations is back-propagation techniques [47,48,49], which allow building attention maps that are self-consistent, consistent in the input domain, and consistent in the space of models [50] in contrast to methods such as sensitivity analysis [51] or class activation maps [52].

One of the emerging approaches is attention guidance [53,54], where the guidance provided, for example, by attention maps highlights relevant regions and suppresses unimportant ones, enabling a better classification. A similar method is based on self-erasing networks that prohibit attention from spreading to unexpected background regions by erasing unwanted areas [55,56]. Researchers developed methods incorporating human expert knowledge into training to avoid the negative influence of medically irrelevant features or artefactual data (Human in the Loop—HIL). HIL AI-based systems [57,58] are systems where a person (e.g., field expert) interacts with a learning system and manually answers questions, selects interesting regions, points out errors in decisions and premises, and simply acts as a guide. In the context of the issues mentioned above, both literature analysis and our research indicate that an effective and reliable approach addressing the problems identified is a form of weakly supervised learning, namely Multiple Instance Learning (MIL) [16,59,60,61,62,63].

MIL-based algorithms enable the generation of attention maps, analysing and indicating the relevance of input data areas for decision making by the neural models. These maps allow experts and end-users to quickly verify the relevance and correctness of the premises, thereby increasing the system’s transparency and reliability. Furthermore, they can guide clinicians on which parts of the analysed image are worth paying particular attention to when making a final diagnosis.

Features of medical datasets such as the relatively small number of annotated data, especially at pixel level (segmentation masks), the large size of the images (e.g., 3518 × 2800 px), and class imbalance make the direct use of commonly used neural models difficult or even impossible. Problems of insufficient or unbalanced data can be addressed to some extent through regularisation techniques, e.g., data augmentation and transfer learning. Due to the large size of the images, the straightforward deployment of the pre-trained deep neural networks such as ResNet or DenseNet [20,64] made available by other researchers is difficult and ineffective because they are designed to operate on a much smaller input size of 224 × 224 px. Recent research concerning the classification of breast cancer in mammograms with deep learning made use of high-resolution images resized to such dimensions as 64 × 64, 192 × 192, 224 × 224, and 256 × 256 px or utilised only a fragment of an image surrounding the region of interest (ROI) with a similar size as previously listed [65,66,67,68]. Reducing images to such a resolution causes the loss of valuable information about breast cancer symptoms [69,70]. In small breasts, it might lead to severely limited model performance caused by abnormalities becoming invisible.

On the other hand, leaving images at their original resolution is impractical and inefficient, as the extracted image features are diluted by a global averaging pooling mechanism that produces a fixed-size representation. Moreover, the images are characterised by a high noise-to-signal ratio—tissues affected by lesions covering a small portion of the examined body part, frequently smaller than 2% of breast volume [71,72]. The task of small object recognition is still a challenge in computer vision, not only in the medical domain.

An analysis of recent achievements in developing deep learning algorithms employed in the medical domain has shown a promising class of supervised learning algorithms called Multiple Instance Learning [60,73]. According to this approach, the whole image is divided into smaller fragments called instances or patches, which form a bag. The bag of instances is given a class based on the image-level annotations. The representations of the individual instances must be aggregated (e.g., using pooling) to obtain a prediction of the entire bag. Successful medical applications of MIL to date include the detection of malignant lesions on histopathology whole-slide imaging (WSI) [61] or the detection of diabetic retinopathy from the eye’s fundus images [74].

## 3. Proposed Methodology

In this paper, we describe and analyse an approach for breast lesion classification and coarse localisation of cancerous tumour lesions, taking advantage of a Clustering-constrained Attention Multiple Instance Learning deep learning classifier. We conducted several comparative analyses of breast lesions’ diagnostic performance based on solely CC images, purely MLO images, analysis of combined CC and MLO images, and multimodal analysis of CC and MLO images, with comprehensive ablation studies showing the influence of hyperparameters on system performance and interpretability. Assessment of mammography always includes two projections, MLO and CC, assessed by the radiologist simultaneously. Lesions should be visible in both projections to confirm the radiological diagnosis. Therefore, both images carry some vital information on cancer characteristics and features. Hence, pooling information from both projections should improve the predictive value of the algorithm.

During the research, we utilised and analysed full-field digital mammography MLO and CC data gathered and annotated at an image level by the Medical University of Gdansk, Poland (MUG dataset). Furthermore, we researched another dataset, the Chinese Mammography Database (CMMD) [75]. The experiments and the obtained results were comprehensively discussed and commented on from the point of view of both medical and ML specialists.

### 3.1. Datasets

The Medical University of Gdańsk (MUG) dataset consists of 1968 full-field digital mammography images (FFDM) from 789 patients (Table 1) undergoing mammography between 2012 and 2022 at the University Clinical Centre in Gdansk, Poland. All cancer images, classified as BI-RADS 4a–c or 5, had a pathological confirmation of the ground truth determined by the histopathological result of the biopsy or post-surgical specimens. The non-cancer group consisted of benign lesions, mainly BI-RADS 2 or 3, and images without any lesions (BI-RADS 1) with a negative follow-up with at least one additional negative mammography or ultrasound scan. In the non-cancer group, several patients were verified by the biopsy if BI-RADS = 3 and in all cases of BI-RADS 4a–c. We included female patients aged 18 or older who underwent mammography of both breasts with radiological reports available and histopathological assessment when needed, all performed in the mentioned clinical centre. Minors, patients with a previous history of breast cancer treatment, any other breast surgery and incomplete data were excluded. Another used dataset named Chinese Mammography Database (CMMD) is publicly available [75] and described in [76]. It consists of 3744 FFDM images acquired from 1775 patients undergoing mammography between 2012 and 2016. The type of benign or malignant tumours was confirmed by biopsy. For each dataset, the entire cohort was divided into two subsets: patients with breast cancer and patients without. There was no overlap of patients between the groups in the MUG dataset, while there were 30 overlapping patients in the CMMD dataset. The clinical details of the patient cohorts are shown in Table 1 and Figure 1.

The study protocol for data collection and storage complies with Polish GDPR. Approval for the study was sought from the Independent Bioethics Committee for Scientific Research at the Medical University of Gdańsk. Due to using anonymised, retrospective data, the requirement for patient-informed consent was waived. All images from the MUG dataset were obtained with a single scanner Siemens Mammomat Inspiration according to the national acquisition protocol. In the case of the CMMD dataset, two scanners were used, the SIEMENS Mammomat Inspiration and the GE Senographe DS, and the image acquisition protocol remains unknown to us. Additional information, including data specifications for both datasets, can be found in Table 2.

The images in MUG and CMMD datasets differ in pixel depth and pixel spacing. Figure 2 illustrates the difference in pixel intensities among images of these two datasets, and the DICOM image overlaying plane is enabled to show a ruler.

### 3.2. Multiple Instance Learning-Based Classification

In deep learning image classification tasks, Multiple Instance Learning aims to synthesise a model that can predict a bag label, e.g., suggestion of medical diagnosis. An image is sliced into small patches with a fixed size and overlapping factors to form a bag. The overlapping factor is a hyperparameter value between 0 and 1 that is selected beforehand, indicating the degree of overlap (0–100%) between adjacent square patches in the x and y axes. When an image is sliced into patches, the process starts from the top left corner of the image. Patches of a fixed square size are cropped with a stride in each axis direction equal to one minus the overlapping factor and then multiplied by the patch size. Due to varying breast sizes, some patches may contain just a tiny part of the breast or completely cover the image background. Thus, we discard those to save computational time. A set of those selected patches is merged into a bag that inherits the whole image label. The deep MIL pipeline consists of a feature extractor in the form of a convolutional neural network that encodes patches into features (low-dimensional soft embeddings), a pooling mechanism to obtain bag embedding, and a classifier [61]. The attention-pooling mechanism is an essential element of the pipeline (Figure 3).

#### 3.2.1. Attention Pooling Mechanism

In many fields, including sensitive and socially relevant ones such as medicine, the explainability of the model is crucial for its trustworthy and, thus, safe use [37,77,78]. The MIL model prediction might be interpreted by skilfully visualising the critical instances discerned by an aggregation operator corresponding to the attention mechanism (Figure 4, Equation (1)). This mechanism functions as a weighted average of instances’ features, whose weights are determined by the dedicated attention neural network that consists of two fully connected layers. The hyperbolic tangent activation function is utilised in one of the layers for proper gradient flow during network training. This construction allows the discovery of (dis)similarities among instances. However, this is inefficient in cases where complex relations must be identified due to approximated linearity for values within the range <−1; 1>. A gated attention pooling involving a sigmoid activation function in the second layer of the neural network is used to alleviate this issue. Afterwards, both layers’ outputs are multiplied element-wise. Mammograms being analysed within the project differ in the number of valuable patches because some patches contain a small amount of breast tissue or negligible image background. Hence, in this case, the softmax function is applied to overcome susceptibility to variable bag size, ensuring that the network weights sum up to one. Obtained weights and extracted features for each instance are utilised in producing the final bag embedding in a weighted average manner. Such generated weights may be employed in attention map production corresponding with image regions that contributed the most to the final prediction. The resolution of such an attention map is directly related to the size of patches and their overlapping factor.

Firstly, each patch undergoes a feature extraction process using a convolutional neural network with the last fully connected layer removed. This results in a set of K patches being encoded in a lower-dimensional feature space $H=\{h_{1},\ldots {,h}_{K}\}$. Afterwards, a pooling mechanism is applied to compute the attention scores. The bag embedding B is finally generated by decoding the patch features and their corresponding attention weights in a weighted sum manner by following MIL pooling:(1)$$B=\sum_{k=1}^{K}a_{k}h_{k}$$
where $a_{k}$ is the *k*-th patch attention weight:(2)$$a_{k}=\frac{exp\left\{w^{T}\left(tanh\left({Vh}_{k}^{T}\right)⊙sigm\left({Uh}_{k}^{T}\right)\right)\right\}}{\sum_{j=1}^{K}exp\left\{w^{T}\left(tanh\left({Vh}_{j}^{T}\right)⊙sigm\left({Uh}_{j}^{T}\right)\right)\right\}}$$
where $w\in R^{L\times 1}$, $V\in R^{L\times M}$ and $U\in R^{L\times M}$ are learnable parameters, and ⊙ is an element-wise multiplication.

Finally, bag embedding is processed by the classification layer, which outputs bag prediction [61]. The bag classifier is a fully connected neural network consisting of a single hidden layer with 256 neurons, followed by an output layer with a single neuron. The output layer applies a sigmoid activation function to produce a probability score between 0.0 and 1.0. A score above 0.5 corresponds to a positive prediction (cancer), while a score below 0.5 corresponds to a negative prediction (non-cancer). Bag level loss corresponds to the entire image prediction generated by the bag classifier by processing the bag embedding. The binary cross-entropy loss is computed by comparing the predicted image-level prediction score with the ground truth image-level label. The importance of each patch in the final prediction depends on computed attention scores, according to Equation (2).

#### 3.2.2. Clustering-Constrained Attention

Due to the relatively small number of positive patches in the positively labelled bags, we took advantage of an exciting approach described in [74], which focuses on grouping and then distinguishing patches as opposite evidence. In this work, the authors employed the MIL system for whole-slide imaging in pathology classification problems, which involved constraining the attention module (Figure 5) by embedding a component that relies on identifying instances of high and low diagnostic values. The operation first sorts the attention scores and then selects the k instances with the largest scores and the k instances with the lowest scores. This process groups them into two clusters: positive and negative evidence, each containing k elements. As in the original CLAM paper, the parameter k was set to 8. The additional task of supervised learning in the network training process is applied for constraining and refining the instance-level feature space by more distinctly separating the features responsible for positive or negative evidence. The instance-level classifier is a fully connected neural network with one hidden layer containing 256 neurons. It is used to classify instance labels for the selected positive and negative evidence patches and, consequently, to compute the instance-level loss. The instance-level loss corresponds to 2000 patches selected during the instance-level clustering, in which pseudo labels were assigned by the positive or negative evidence cluster. The instance-level prediction scores for each selected patch are compared against their corresponding pseudo-cluster labels using the cross-entropy loss. In that fashion, we could exploit more concentrated attention weight values to produce attention maps focused to a greater extent.

### 3.3. Implementation Details

#### 3.3.1. Data Preprocessing for Machine Learning Model

The pixel intensity values of each single-channel image were linearly scaled to the 0–1 range and then repeated to 3 channels owing to CNNs, which perform image processing on multi-channelled images. Afterwards, the image was divided into fixed-size, square-shaped tiles, called instances or patches, to compose a set called a bag. The composition of the bag was made by selecting instances containing at least 75% non-zero pixels to avoid including the background image and thus reducing computational time. Figure 6 shows an example of a patch-wise image division and selecting tiles that meet the condition of having more than 75% non-zero pixels. The sub-image on the far right illustrates an example of a single patch and its two neighbouring patches that overlap by 50% (overlapping factor: 0.5) in both the x and y directions.

The experiments tested three different patch sizes and three different overlap ratios (128 px, 224 px, 448 px for patch sizes and 0.00, 0.25, 0.50 for overlap ratios), and every combination is shown in Table 3. Furthermore, the study examined the full resolution of the images and images resized by half. Additionally, the information hidden within the full-field digital mammography MLO and CC images individually and jointly were explored through data fusion in utilising both views of the breast (CC and MLO)—multimodal view.

Initial works on the MUG dataset identified the need for an additional preprocessing technique since the attention module was, in some cases, overly attentive to the edges of the image. This was caused by a noise in the form of a high pixel-intensity line located at the edge of the image, which was related to collimator misalignment [79]. Each breast image was shifted in the chest wall direction by an a priori-selected number of pixels to eliminate this artefact.

#### 3.3.2. Data Augmentation

Due to limited training data, several data augmentation techniques were applied, including rotating, flipping, and resized cropping. Each transformation was performed at the instance level. Augmentations were applied sequentially in a commonly used online approach—images were augmented on the fly. The angle of rotation was randomly selected from 4 predefined values. In the case of Resized Crop, the cropped portion of the instance was always resized back to the previous instance size. The amount of data generated depends on probabilities, possible choices, and the number of training epochs. Since we utilised a random oversampling method, the number of new samples generated for each class was approximately equal. The exact parameters are shown in Table 4. Data augmentation was applied only for the training data, while validation and test data remained untransformed apart from a pixel intensity scaling and an image channel expansion.

#### 3.3.3. Models Used

The input images of the shape H × W × C (H—height; W—width; C—channels) are first converted into bag form of the shape N × H′ × W′ × C (N—number of patches; H′—patch height; W′—patch width; C—channels), and then patch features are extracted by shared CNN. We used feature extractors pre-trained on ImageNet [80] in the form of CNNs such as ResNet-18, ResNet-34, ResNet-50, and EfficientNet-B0 [20,21]. The classification head was removed from each network to access the extracted features. The feature extractors being utilised are listed and compared in Table 5.

In the attention-based module, the feature space undergoes dimensionality reduction using an FC layer with a ReLU activation function outputting 256 features describing each patch in the bag. Afterwards, an attention pooling mechanism consisting of 2 FC layers is applied to obtain bag embedding, as described in Section 3.2.1. In each FC layer contained in the attention-based model, a dropout regularisation technique was applied with a probability of zeroing a neuron equal to 0.25 during the training. At the last stage, produced bag embedding is classified by the FC layer with 256 neurons and sigmoid activation function, resulting in whole image prediction. Figure 7 illustrates the general classification pipeline.

#### 3.3.4. Training Parameters

In each experiment, the models are trained for at least 20 epochs by introducing the early stopping method (20 epochs patience) and up to 100 epochs with the learning rate lr=0.001, linearly decayed every ten epochs by a constant factor gamma=0.9, L2 regularisation (weight decay) wd=0.001 is used, and momentum=0.9 with stochastic gradient descent (SGD) as the optimiser. Due to GPU memory constraints, we set the batch size to 1, meaning that only one case at a time was processed. However, our shared feature extractor CNN perceives the input bag as a shape with a batch size equal to the number of instances in the bag. Therefore, we decided to freeze BatchNorm2d layers located in the pre-trained CNN. Additionally, we applied a gradient accumulation technique with 8 sample steps during the training. This solution imitated a larger batch size, and the sequentially accumulated results were used to update network weights once every eight samples. In the last network update during each epoch, this value may be different if the number of data samples is not divisible by eight. During the training, we used the binary cross-entropy (BCE) loss function for the bag classification task and the categorical cross-entropy (CCE) loss function for the constraining instance-level feature space task. The total loss was computed as the sum of both losses, with a scaling factor of 0.7 for the former and 0.3 for the latter.

The 5-fold cross-validation (CV) was used to estimate general model performance. The dataset was randomly divided into training, validation, and test sets (70%, 10%, and 20%, respectively) considering the uniqueness of the patient’s presence in the subsets. The validation set was obtained by further splitting the training set (4 folds) and used for monitoring the model’s performance during training and model selection. In contrast, the test set (1 fold) was held out to evaluate selected models. The number of patients for the training, validation, and test set in each fold from the MUG dataset is shown in Table 6.

Due to the class imbalance, we applied a weighted random oversampling method (ROS) during the training to enable the algorithm to provide predictions more reliably [81]. This approach works well due to assigning a greater probability of taking a sample from the minority class proportional to the imbalance. In combination with many variants of data augmentation applied to each instance separately, the diversity of cases alleviates the disadvantages of ROS.

#### 3.3.5. Software and Hardware

All code was implemented in Python (v3.8) using PyTorch (v1.10.1) framework and Scikit-learn (v0.24.2) as the primary deep learning packages. Random number generators were initialised with a seed equal to 42. Experiments were conducted on two workstations, one working under Ubuntu 18.04 with Intel(R) i9-10920X (3.50 GHz) CPU and NVIDIA GeForce GTX 1080 Ti (12 GB) GPU, and the other working under Linux Mint 20.3 with AMD Ryzen 9 3950X (3.50 GHz) CPU and NVIDIA GeForce RTX 3090 (24 GB) GPU.

## 4. Results

Five metrics were employed to assess the performance of each model: Accuracy (3), Precision (4), Recall (5), F1-score (6), and receiver operating characteristic area under the curve (ROC AUC), where *TP* is true positive, *TN* is true negative, *FP* is false positive, and *FN* is false negative.
(3)$$Accuracy=\frac{TP+TN}{TP+TN+FP+FN}$$
(4)$$Precision=\frac{TP}{TP+FP}$$
(5)$$Recall=\frac{TP}{TP+FN}$$
(6)$$F1\;score=\frac{2TP}{2TP+FP+FN}$$

### 4.1. Ablation Study

We investigated the contribution of the undertaken regularisation techniques described in Section 3.3 to the system performance. The details of the experiments carried out are as follows: patient IDs were randomly split into the train, validation, and test sets (75%, 12.5%, and 12.5%, respectively), with ResNet-18 as the feature extractor, fixed base parameters for a patch size of 224 × 224 px, and an overlapping factor of 0.5. Table 7 summarises the detailed metrics results for the ablation study, allowing some conclusions to be drawn. Resizing an image by half its original size while keeping the patch size fixed significantly boosts all the metrics (about six p.p). As expected, data augmentation methods led to enhanced performance of the model. The use of dropout regularisation slightly worsened most metrics. The application of a weighted sampler resulted in minor refinement. However, by combining every single previously mentioned change with the accumulating gradient, allowing us to imitate a larger batch size, the outcome was significantly upgraded.

### 4.2. A Single-View Breast Cancer Classification

In the second set of experiments, we trained the system using a single-view breast image (CC or MLO). We used three different data pools during the training and evaluation process to comparatively analyse breast cancer diagnosis performance, and the pools are as follows:Only CC view images (MLO images were not used);Only MLO view images (CC images were not used);Both CC and MLO view images (each view treated as a separate case).

For the a priori-selected patch size, overlapping factor, and CNN for extracting features, which allowed us to utilise available GPU memory fully, we obtained results gathered in Table 8 and shown in the boxplot (Figure A1—Appendix A). The 5-fold cross-validation, as specified in Section 3.3.4, was performed to generate more objective results.

### 4.3. A Multimodal-View Breast Cancer Classification

As mentioned above, both CC and MLO projections contain valuable information indicating the presence and nature of cancerous lesions. We concatenated both views during the training to investigate the effectiveness of such an approach using the proposed CAD system. This method improved metric scores compared to the cases when using the image views separately during the training. As before, the following experiments were conducted with ResNet-18 backbone for two image resolutions (whole and halved), and more variants were tested for patch sizes and overlap ratios. The obtained results are presented in Table 9 and shown in the boxplot (Figure A2—Appendix A).

### 4.4. Comparison with Other CNN Pre-Trained Models for Multimodal-View Breast Cancer Classification

The method described in Section 4.3 was assessed on popular CNNs other than ResNet-18, which include ResNet-34, ResNet-50, and EfficientNet-B0. The results of performed tests are shown in Table 10 and in the boxplot (Figure A3—Appendix A). When comparing the AUC-ROC value, ResNet-34 was the best, while the performance of the ResNet-50 was noticeably worse.

### 4.5. Identification and Influence of Artefacts on Classification—Advantages of the MIL Explainability Mechanism

After training the model with attention weights, the breast regions that contributed the most to the model’s decision can be visualised. Received attention maps might be consulted with domain specialists. Consequently, it is possible to verify if the model works appropriately, not only through obtained metrics, which may be misleading. The first model trainings were conducted on multimodal-view, resized images with no regularisation mechanisms at all, the resulting metrics were surprisingly high, and the model required a small number of epochs to complete the training. Introducing more regularisation techniques into the training did not improve the results. It led us to analyse the results in each fold in great detail, including an analysis of the generated attentional maps. A comprehensive analysis of the attention maps revealed that the most attended regions were those at the edge of the image near the chest wall. As it turned out, the reason for that was the presence of “an edge” artefact in some images, examples of which are shown in Figure 8.

Furthermore, there was a high correlation in the data between images affected by artefacts and images classified as cancerous (ground truth). Therefore, the system erroneously inferred that the presence of an “edge” artefact was a premise for classifying such images as cancerous. To confirm this observation, we eliminated the “edge” artefact from all images within the MUG dataset during preprocessing. This rationalised the learning process, which could be observed both in the metrics and the learning time. Furthermore, the accompanying attentional maps analysis showed that the system started paying attention to patches with medical diagnostic value instead of those lying at the images’ edge. A comparison of the results obtained using biased and preprocessed images (by removing a vertical bar of 20 pixels width from each image) is shown in Table 11 and in the boxplot (Figure A4—Appendix A).

### 4.6. Interpretable Decisions—Attention Heatmaps

Attention weights obtained as a result of the classification process might be used for visualisation purposes. First, an empty image (matrix) is created with the same shape as the original image. Then, the attention score of each patch is added to the corresponding square area of the image determined by the patch’s coordinates. The pixel values are then averaged based on the number of overlaid patches that contain that pixel. Finally, min–max scaling is applied to enable proper visualisation of shades of grey for human perception. The values in the heat maps depend on the patch’s diagnostic value. Namely, the brighter the area on the map is, the more critical it is. Higher heat map resolution can be achieved by increasing the overlap rate while using a patch size suitable for the trained model.

The usefulness of using interpretable methods in the form of heat maps can be seen by comparing images a–e with image f from Figure 9. It can be observed that the correct prediction of cancer in f is caused by dataset-specific image artefacts, which mainly occurred in cancer cases. Hence, the attention module learned to focus on patches taken near the edge of the image.

### 4.7. Summary of Results with Optimal Threshold Selection

The default value for the threshold in binary classification with class labels 0 (non-cancer) and 1 (cancer) is 0.5. This means the predicted class is cancer when the score exceeds 0.5. When lower than 0.5, the prediction is non-cancer. This section presents the results obtained after optimising the threshold by maximising the Youden index (Table 12 and Figure 10).

### 4.8. Evaluation with the CMMD Dataset

Since the CMMD dataset does not have a predetermined test set, only the general performance of the model might be compared to other research papers. For this reason, we employed a 5-fold cross-validation technique taking into account patient-wise breakdowns. Additionally, we used transfer learning of the knowledge gained by the model learned on the MUG dataset described in Section 4.3 to solve a closely related problem on the CMMD dataset, where the main difference is the image domain and presence of the lesion subtypes. Table 13 presents the obtained results next to the main MIL parameters and shows if the concatenated MLO and CC breast view image was resized and whether the transfer learning was applied. Sample attention maps of correct and incorrect predictions are presented in Figure 11.

## 5. Discussion

Acquisition of pixel-level annotated medical imaging data is very time-consuming and requires the involvement of specialist radiologists. Therefore, using weakly supervised learning methods to deal with data annotated just on the image level is practical. Multiple Instance Learning-based models are well suited for medical classification problems due to the possibility to efficiently handle large-resolution images by patching the whole image into smaller fragments. Additionally, due to the use of an attention-based pooling mechanism, it is possible to visualise the reasoning process of the model. Thus, the location of the lesion may be provided without the need to use pixel-level annotations during the training.

Furthermore, we showed a practical example that ensuring the availability of insight into the decision-making process of deep learning algorithms is essential. Especially in the analysis of medical images, model performance can be significantly biased by the presence of medically unimportant artefacts that may be invisible at first glance. This has a positive effect on the radiologists’ confidence in the decision taken by the system and allows machine learning specialists to search for biases and improve the system’s performance.

Model hyperparameters and employed regularisation techniques significantly affect the performance. In particular, imitating larger batch size through gradient accumulation and counteracting data imbalance with a weighted sampler combined with severe data augmentation applied on each patch help face the lack and imbalance of data problems. Parameters of MIL in the image domain, such as overlapping factor and patch size, seem crucial in tuning, but available GPU memory makes it hardware-dependent. We found that the best results were obtained for a patch size of 224 × 224 pixels, and this is probably caused by the fact that it is the standard input size of used feature extractor CNNs. On the other hand, such resolution also seems suitable for detecting and distinguishing relevant cancer features. With larger overlapping factors, more patches form a bag, which may affect the performance positively or negatively. Gathering more differing image fragments artificially increases available data size, and there is a bigger probability of, at best, covering tumour regions. However, for large amounts of patches, the attention weights might not be adequately learned and thus diluted. The best metrics were obtained for overlapping of 25% in each axis, which supports the above reasoning.

In light of the research we conducted, it is evident that combining CC and MLO views into the same bag substantially improves the model performance by taking advantage of both projections (about a 6 p.p. increase compared with the single CC or MLO). Using only single-view CC or MLO images resulted in similar metrics with a slight advantage in favour of the MLO view. This is consistent with clinical practice, where it provides more diagnostic value and better assessment of a larger portion of the mammary gland, especially the tail of Spence. However, the model might be harder to learn due to the presence of patches containing pectoral muscle regions, which disturb the bag with its high-intensity pixel patches.

Every used CNN has many learnable parameters sufficient for data scarcity problems. This might be why models performed slightly better when extracting lower-dimensional feature spaces because it was simpler to learn attention networks. Experiments using different CNNs as feature extractors performed somewhat better for ResNet18 and ResNet34, producing 512-dimensional feature space, less than ResNet50 and EfficientNet-B0.

In addition, we verified our approach on the Chinese Mammography Database (CMMD) dataset. It is a relatively new publicly available breast cancer dataset, and not many papers refer to it yet. Therefore, our work may be a valuable reference for the other researchers who will analyse this dataset in their studies. The CMMD does not have a predetermined test set, so due to the potential differences in test conditions, a direct comparison of methods is impossible, and only overall performance can be compared. Stadnick et al. [82] tested five open-sourced screening mammography classifiers on multiple public datasets, including CMMD. The AUC-ROC results presented by [82] range between 0.449 and 0.831, and our 5-fold cross-validation resulted in an average AUC-ROC of 0.848, outperforming each of the seven reported methods for CMMD. Walsh, Tardy in [83] trained and tested the CMMD dataset using different techniques for class imbalance. The reported AUC-ROC results obtained by [83] are in the range of 0.681–0.727. Our averaged results without transfer learning AUC-ROC of 0.772 and with transfer learning AUC-ROC of 0.848 are better than any of the four experimental procedures used.

Finally, as shown in this paper, assembling appropriate deep learning algorithms for cancer image analysis and classification requires a lot of domain knowledge and close cooperation between medical and AI teams. Approaches involving end-users in all stages, such as attention maps in the form of heatmaps, allow a better understanding of causes for system predictions. In some cases, it might enable the identification of erroneous predictions based on the wrong premises, which can cause many problems for patients and their doctors using the system.

## 6. Conclusions

This paper presented the approach for breast cancer diagnosis based on the Clustering-constrained Attention Multiple Instance Learning (CLAM) algorithm. The approach has the advantage of training on data labelled just at the image level. Moreover, it allows effective diagnosis even with a relatively small number of high-resolution images.

We have experimentally confirmed that, in line with medical knowledge, the best results can be obtained from a joint (multimodal) analysis of images from the CC and MLO projections.

We comprehensively analysed the impact of the system’s hyperparameters and selected regularisation methods on its efficiency. Ablation study results were carefully examined, and the CAD system was finally synthesised on this basis.

We also validated the system on a publicly available CMMD database, which enabled us to compare the results with those reported by other researchers. It should be emphasised that these databases differed significantly in terms of the number of images, pixel intensity levels, and subtypes of existing lesions. The results achieved surpass those reported in the literature.

Furthermore, we showed that a system trained on one dataset might be successfully employed to train a system on a terminal dataset more effectively in the transfer learning process.

It was confirmed that, with appropriately chosen hyperparameters, the CLAM algorithm’s attentional maps indicate the features most relevant to the algorithm in the images. Notably, the indicated spots are indeed medically relevant. Our valuable discovery is the identification of an artefact in the analysed dataset, previously unnoticed by the medical experts and clinically irrelevant, which significantly distorted the results of the CAD system.

The AI-based system developed is effective both on the database for which it was prepared and on another one—after additional training. Furthermore, the visualisation in the form of attentional maps of the features on which the diagnosis was founded makes the system more trustworthy for the end users, i.e., the medical specialists. It also allows ML specialists to discover biases and artefacts in the datasets and flaws in the algorithm itself more efficiently.

We are convinced that developing effective and trustworthy AI-based computer-aided diagnosis systems is only possible through the synergetic cooperation of specialists from different fields—AI, ML, and decision support specialists on the one hand and radiologists, oncologists, surgeons, and medical physicists on the other.

## Figures and Tables

**Figure 1 cancers-15-02704-f001:**
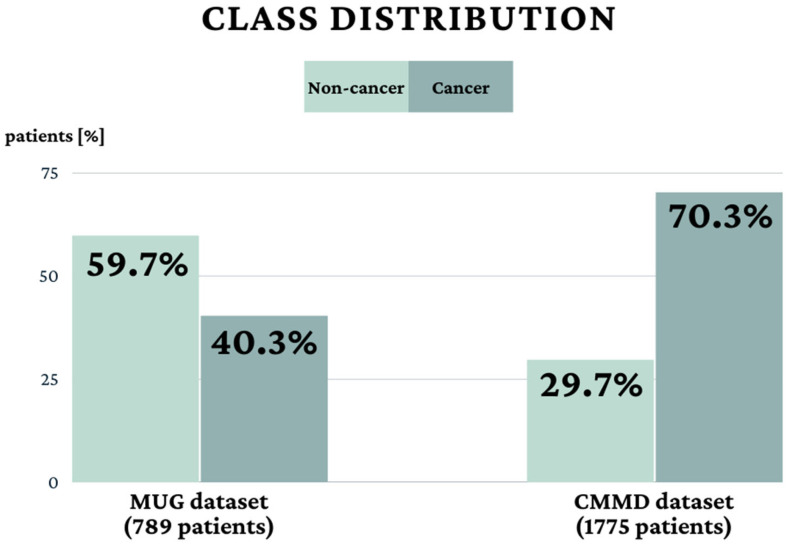
Patients class distribution in MUG and CMMD datasets.

**Figure 2 cancers-15-02704-f002:**
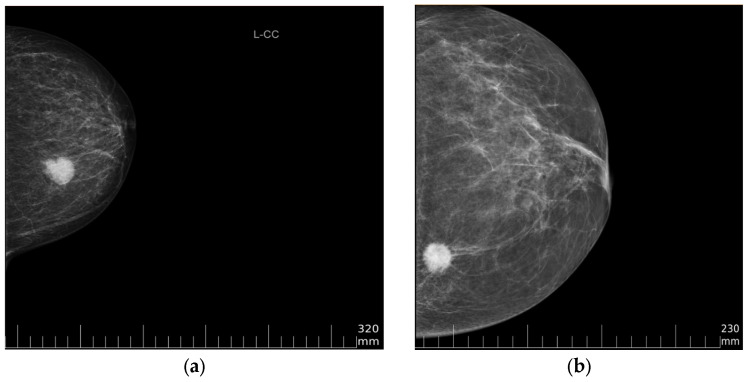
Data sample from (**a**) MUG dataset and (**b**) CMMD dataset.

**Figure 3 cancers-15-02704-f003:**
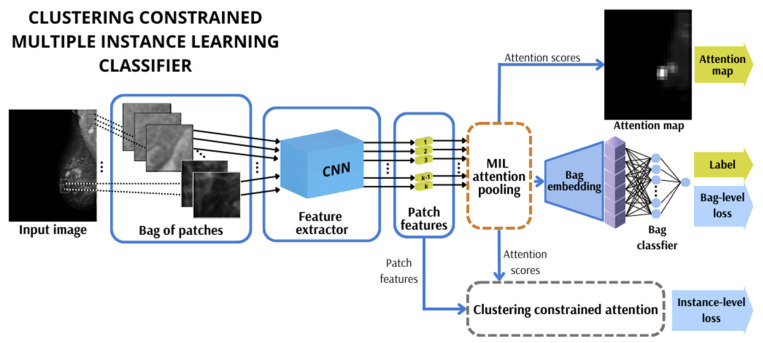
The general pipeline of the Clustering-constrained Attention Multiple Instance Learning-based classification scheme.

**Figure 4 cancers-15-02704-f004:**
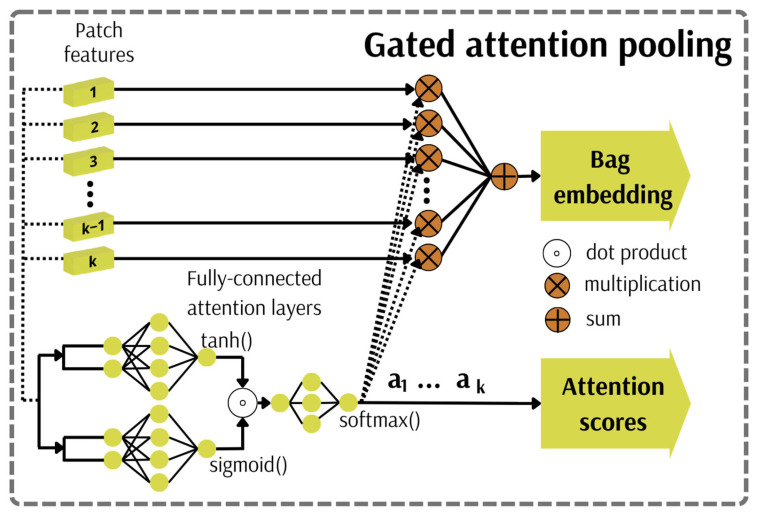
Attention pooling mechanism.

**Figure 5 cancers-15-02704-f005:**
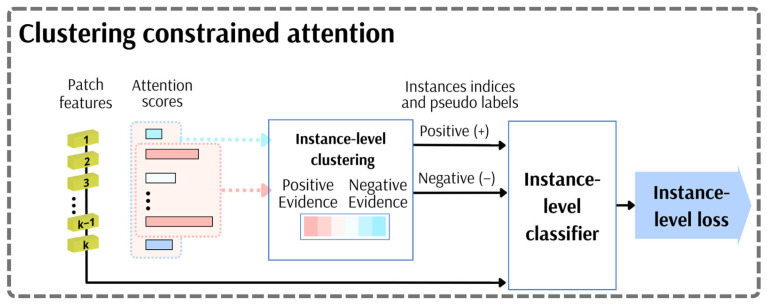
Attention-constraining module.

**Figure 6 cancers-15-02704-f006:**
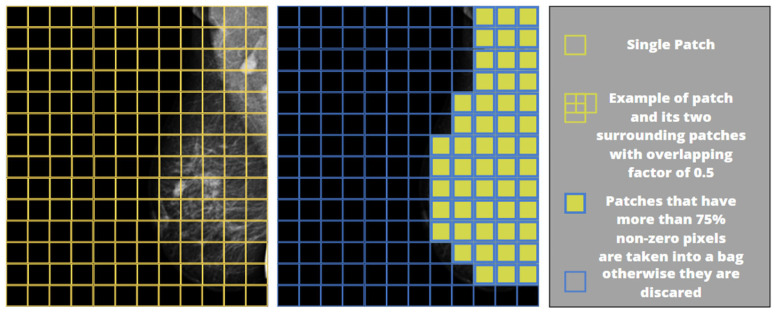
Selection of image patches to form a bag.

**Figure 7 cancers-15-02704-f007:**
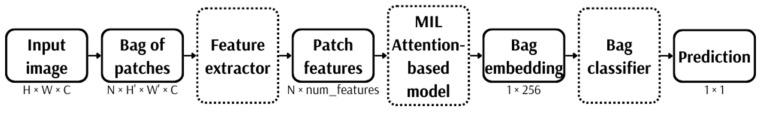
General attention-based MIL classification pipeline.

**Figure 8 cancers-15-02704-f008:**
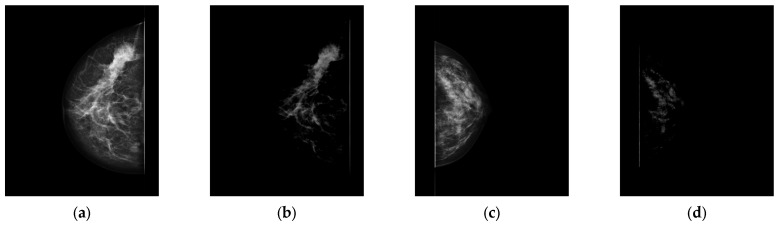
Mammographic images of the breast with visible artefacts: (**a**,**c**) default view, (**b**,**d**) view after application of windowing.

**Figure 9 cancers-15-02704-f009:**
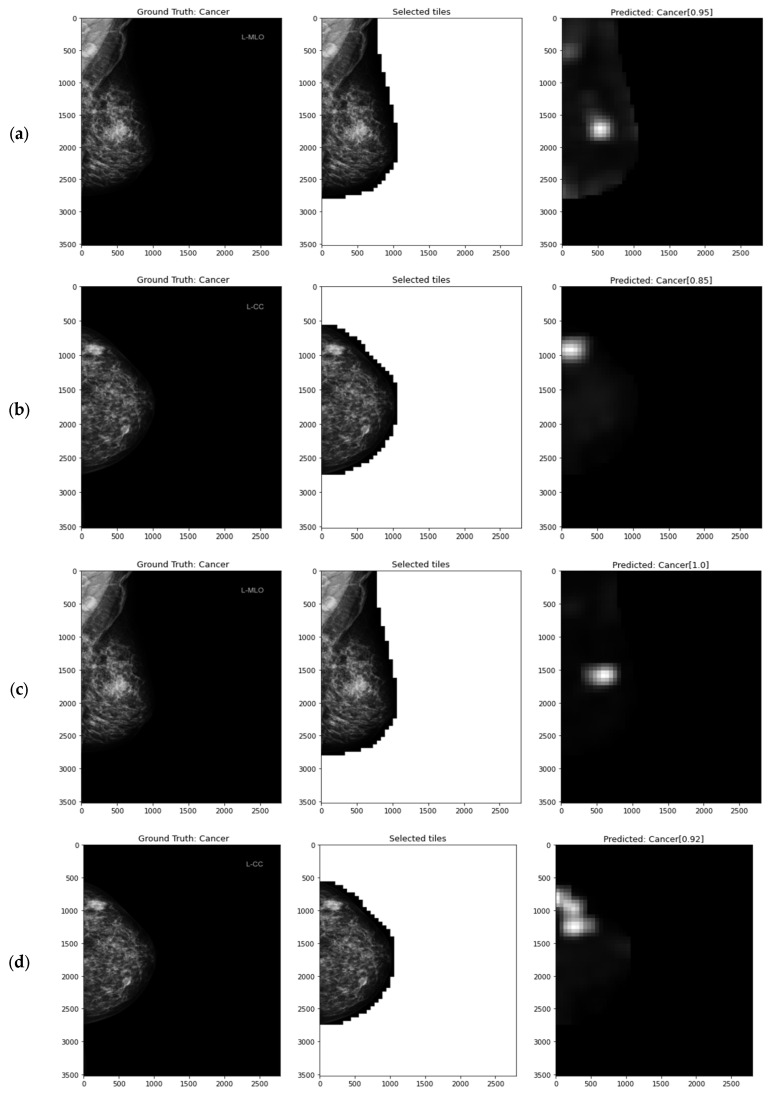
Left: whole FFDM image and ground truth label; middle: selected image tiles (patches); right: class prediction, score, and produced heatmap for (**a**) single-view approach using MLO view only, (**b**) single-view approach using CC view only, (**c**,**d**) single-view approach using MLO and CC views separately, respectively, (**e**) multimodal-view approach using MLO and CC views simultaneously, (**f**) same as (**e**) but with image artefacts not removed.

**Figure 10 cancers-15-02704-f010:**
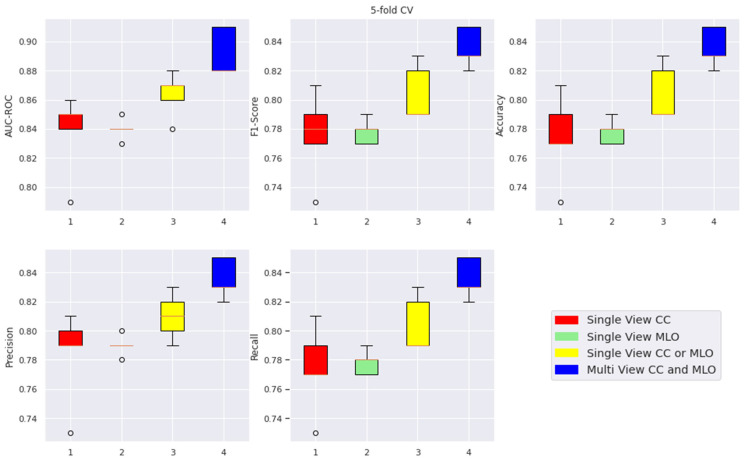
Boxplots of 5-fold cross-validation metrics comparison with optimised thresholds.

**Figure 11 cancers-15-02704-f011:**
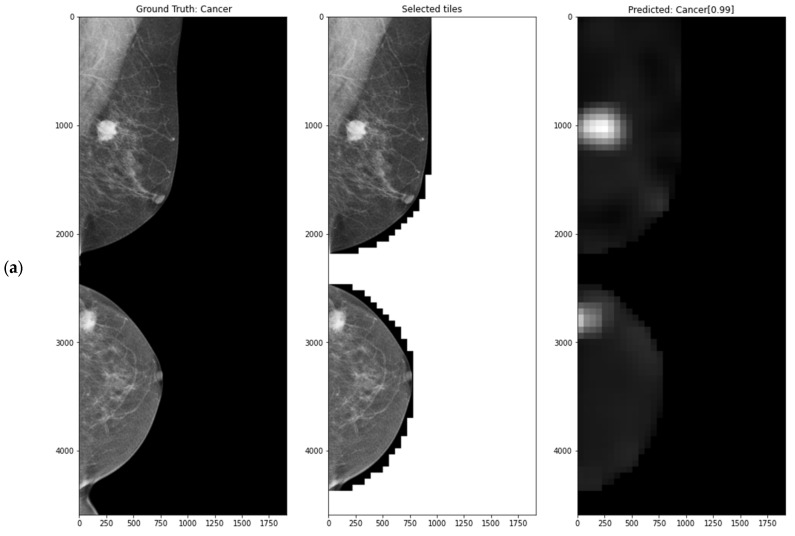
Left: whole FFDM image and ground truth label; middle: selected image tiles (patches); right: class prediction, score, and produced heatmap for (**a**–**c**) true positive and (**d**) false negative prediction.

**Table 1 cancers-15-02704-t001:** Clinical details of the patient cohort.

	MUG Dataset		CMMD Dataset	
	Non-Cancer	Cancer	Non-Cancer	Cancer
Number of patients	392	397	465 + 30 *	1280 + 30 *
Median Age (Min–Max)	51 (21–86)	57 (29–88)	43 (17–84)	49 (21–87)
Number of images of the left breast **	582	412	562	1414
Number of images of the right breast **	592	382	546	1222

* Both groups contain the same 30 patients with bilateral images with benign findings in one breast and malignant in the other. ** CC images and MLO images in total.

**Table 2 cancers-15-02704-t002:** Specification of data.

	MUG Dataset	CMMD Dataset
Type of data	DICOM files—full-field digital mammography craniocaudal (CC) and mediolateral oblique (MLO) images Excel file—age, pathological ground truth	DICOM files—full-field digital mammography craniocaudal (CC) and mediolateral oblique (MLO) images Excel file—age, pathological ground truth
Image acquisition protocol	National acquisition protocol	Unknown
Image acquisition	SIEMENS Mammomat Inspiration	SIEMENS Mammomat Inspiration, GE Senographe DS
Image resolution	3518 × 2800 pixels	2294 × 1914 pixels
Pixel spacing	0.085 mm	0.094 mm
Pixel depth	12-bit (0–4095)	8-bit (0–255)
Data source location	University Clinical Center in Gdańsk, Poland	SunYat-sen University Cancer Center in Guangzhou, Nanhai Affiliated Hospital of Southern Medical University in Fushan, China

**Table 3 cancers-15-02704-t003:** Experimental parameters.

Experiment	Image Size (px)	Patch Size (px)	Overlap (-)
Single view	3518 × 2800 *	224 **	0.50
Single view with resize	1759 × 1400	224	0.50
Multimodal view	7036 × 2800	128	0.50
		224	0.00, 0.25, 0.50
		448	0.50
Multimodal view with resize	3518 × 1400	128	0.50
		224	0.00, 0.25, 0.50
		448	0.50

* Without resizing, one pixel corresponds to the breast area size of 0.085 × 0.085 mm. ** Thus, the patch in size of 224 × 224 px = 19.04 × 19.04 mm.

**Table 4 cancers-15-02704-t004:** Instance/patch-level augmentations.

No.	Augmentation Methods	Parameters
1	Rotate	Angle = (0°, 90°, 180°, 270°), Probability = 1
2	Horizontal Flip	Probability = 0.5
3	Vertical Flip	Probability = 0.5
4	Resized Crop	Scale range [0.8–1.0], Probability = 1

**Table 5 cancers-15-02704-t005:** Comparison of feature extractors.

Feature Extractor	Number of Parameters	Feature Space Dimension
ResNet-18	11.7 M	512
ResNet-34	21.8 M	512
ResNet-50	25.6 M	2048
EfficientNet-B0	5.3 M	1280

**Table 6 cancers-15-02704-t006:** The number of patients split between three sets within five subsequent folds from the MUG dataset.

Fold		Training Set	Validation Set	Test Set
1	(non-cancer)	280	40	72
	(cancer)	272	39	86
2	(non-cancer)	270	38	84
	(cancer)	282	41	74
3	(non-cancer)	274	37	81
	(cancer)	278	42	77
4	(non-cancer)	278	39	75
	(cancer)	274	40	83
5	(non-cancer)	281	31	80
	(cancer)	272	48	77

**Table 7 cancers-15-02704-t007:** Ablation study results—the influence of employed regularisation techniques on system performance. (✔—applied; ✖—not applied).

Resize	Transforms	Dropout	Weighted Sampler	Gradient Accumulation	AUC-ROC (-)	F1-Score (%]	Accuracy (%)	Precision (%)	Recall (%)
✖	✖	✖	✖	✖	0.81	71.0	72.0	74.0	72.0
✔	✖	✖	✖	✖	0.87	78.0	78.0	79.0	78.0
✔	✔	✖	✖	✖	0.87	81.0	81.0	81.0	81.0
✔	✔	✔	✖	✖	0.87	79.0	79.0	80.0	79.0
✔	✔	✔	✔	✖	0.89	80.0	80.0	80.0	80.0
✔	✔	✔	✔	✔	**0.92**	**88.0**	**88.0**	**88.0**	**88.0**

**Table 8 cancers-15-02704-t008:** Single-view classification results for 5-fold cross-validation.

Feature Extractor	View	Image Size (px)	Patch Size (px)	Overlap (-)	AUC-ROC (-)	F1-Score (%)	Accuracy (%)	Precision (%)	Recall (%)
ResNet-18	CC	3518 × 2800	224	0.5	0.792 ± 0.026	72.4 ± 2.5	72.6 ± 2.8	72.8 ± 2.7	72.6 ± 2.8
		1759 × 1400			0.838 ± 0.025	76.2 ± 3.5	76.2 ± 3.7	77.2 ± 3.8	76.2 ± 3.7
	MLO	3518 × 2800			0.804 ± 0.053	72.8 ± 4.9	72.8 ± 4.4	74.0 ± 4.8	72.8 ± 4.4
		1759 × 1400			0.840 ± 0.006	76.0 ± 1.4	76.0 ± 1.6	76.4 ± 1.4	76.0 ± 1.7
	CC or MLO	3518 × 2800			0.822 ± 0.021	75.6 ± 2.8	75.8 ± 3.0	75.6 ± 2.9	75.8 ± 3.0
		1759 × 1400			**0.864 ± 0.014**	**78.4 ± 0.4**	**78.4 ± 0.5**	**79.4 ± 0.8**	**78.6 ± 0.5**

**Table 9 cancers-15-02704-t009:** Multimodal-view classification results for 5-fold cross-validation.

Feature Extractor	View	Image Size (px)	Patch Size (px)	Overlap (-)	AUC-ROC (-)	F1-Score (%)	Accuracy (%)	Precision (%)	Recall (%)
ResNet-18	CC and MLO	7036 × 2800	128	0.5	0.846 ± 0.027	75.6 ± 4.8	76.0 ± 4.1	76.6 ± 4.4	76.0 ± 4.1
			224	0	0.828 ± 0.033	75.8 ± 3.5	76.0 ± 3.5	75.8 ± 3.5	76.0 ± 3.5
			224	0.25	0.850 ± 0.030	76.4 ± 4.1	76.6 ± 3.8	77.6 ± 4.3	76.6 ± 3.8
			224	0.5	0.844 ± 0.026	77.0 ± 3.8	77.0 ± 3.8	77.4 ± 3.9	77.0 ± 3.8
			448	0.5	0.828 ± 0.040	76.2 ± 5.7	76.6 ± 5.2	77.0 ± 5.0	76.6 ± 5.2
		3518 × 1400	128	0.5	0.894 ± 0.022	79.8 ± 3.2	79.8 ± 2.6	80.2 ± 3.4	79.8 ± 2.6
			224	0	0.892 ± 0.012	79.8 ± 1.7	80.0 ± 2.1	80.4 ± 2.1	80.0 ± 2.1
			224	0.25	**0.896 ± 0.017**	**81.8 ± 3.2**	**81.6 ± 3.2**	**82.4 ± 3.3**	**81.6 ± 3.2**
			224	0.5	0.892 ± 0.015	81.6 ± 1.9	81.6 ± 1.9	82.0 ± 1.7	81.6 ± 1.9
			448	0.5	0.866 ± 0.039	77.2 ± 5.4	77.6 ± 4.7	78.2 ± 4.6	77.6 ± 4.7

**Table 10 cancers-15-02704-t010:** Multimodal-view classification results for 5-fold cross-validation with different backbone CNNs.

Feature Extractor	View	Image Size (px)	Patch Size (px)	Overlap (-)	AUC-ROC (-)	F1-Score (%)	Accuracy (%)	Precision (%)	Recall (%)
ResNet-18	CC and MLO	3518 × 1400	224	0.5	0.892 ± 0.015	81.6 ± 1.9	81.6 ± 1.9	82.0 ± 1.7	81.6 ± 1.9
ResNet-34					**0.892 ± 0.023**	**82.2 ± 2.6**	**82.2 ± 2.6**	**82.4 ± 2.4**	**82.2 ± 2.6**
ResNet-50					0.878 ± 0.013	79.8 ± 3.5	80.2 ± 3.3	80.8 ± 2.2	80.2 ± 3.3
EfficientNet-B0					0.890 ± 0.019	81.4 ± 1.6	81.6 ± 1.4	82.2 ± 1.2	81.6 ± 1.4

**Table 11 cancers-15-02704-t011:** Metrics for 5-fold cross-validation with and without the presence of image artefacts.

Edge Artefacts	AUC-ROC (-)	F1-Score (%)	Accuracy (%)	Precision (%)	Recall (%)
Absent	0.846 ± 0.027	76.0 ± 3.7	76.4 ± 4.0	76.4 ± 4.0	76.4 ± 4.0
Present	0.866 ± 0.024	82.8 ± 2.0	82.8 ± 2.0	82.8 ± 2.0	82.8 ± 2.0

**Table 12 cancers-15-02704-t012:** Fivefold cross-validation results comparison with optimised thresholds.

Feature Extractor	View	Image Size (px)	Patch Size (px)	Overlap (-)	AUC-ROC (-)	F1-Score (%)	Accuracy (%)	Precision (%)	Recall (%)
ResNet-18	CC	1759 × 1400	224	0.5	0.838 ± 0.025	77.6 ± 2.7	77.4 ± 2.7	78.4 ± 2.8	77.4 ± 2.7
	MLO				0.840 ± 0.006	77.8 ± 0.7	77.8 ± 0.7	79.0 ± 0.6	77.8 ± 0.7
	CC or MLO				0.864 ± 0.014	80.4 ± 1.7	80.4 ± 1.7	81.0 ± 1.4	80.4 ± 1.7
	CC and MLO	3518 × 1400			**0.892 ± 0.015**	**83.6 ± 1.2**	**83.6 ± 1.2**	**83.6 ± 1.2**	**83.6 ± 1.2**

**Table 13 cancers-15-02704-t013:** Fivefold cross-validation results on the CMMD dataset.

Transfer Learning	View	Image Size (px)	Patch Size (px)	Overlap (-)	AUC-ROC (-)	F1-Score (%)	Accuracy (%)	Precision (%)	Recall (%)
Not applied	CC and MLO	2294 × 957	224	0.5	0.786 ± 0.050	72.4 ± 3.3	72.0 ± 3.1	75.4 ± 3.9	72.0 ± 3.1
		4588 × 1914			0.772 ± 0.019	73.6 ± 0.5	73.0 ± 1.4	74.4 ± 1.0	73.0 ± 1.4
Applied		2294 × 957			0.790 ± 0.013	73.6 ± 1.5	74.4 ± 1.2	73.8 ± 1.6	74.4 ± 1.2
		4588 × 1914			**0.848 ± 0.015**	**78.6 ± 2.0**	**78.4 ± 1.9**	**78.8 ± 2.0**	**78.4 ± 1.9**

## Data Availability

The Chinese Mammography Database (CMMD) is available at the Cancer Imaging Archive: https://wiki.cancerimagingarchive.net/pages/viewpage.action?pageId=70230508#70230508e1942f5ac79345328ce7713bd023e3cc (accessed on 15 March 2023). The private MUG dataset presented in this study is available on request from the corresponding author.

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
