# Peer review of "Attention-Based Deep Learning System for Classification of Breast Lesions—Multimodal, Weakly Supervised Approach"

_cancers, 2023, doi:10.3390/cancers15102704_

Round 1
Reviewer 1 Report
Dear authors:
1) explain "overlapping factor" and in what way you are applying it. Figure 6 could be used to illustrate the concept. The term appears a few times in the text and each time seems to be computed in a different way.
2) section 3.3.2 data augmentation. Did you use this technique at all?
How many new samples were generated ? How many of each class ?
What is the source data?
Were all the augmented samples result of all operations in Table 4 ?
(operations 2 and 3 are applied with some probability)
3) Figure 3. Once you get the patch features the proposed system becomes a too black black box. Figure 4 MIL attention score. What is the target value of the 2 fully connected NN ?
How do you perform bag embedding ?
Attention scores go to process in Figure 5. How many clusters ?
The instance level classifier is a .....?
What is instance level loss ?
Bag level loss ??
The attention map is shown in figure 7. A heat map is an intensity pixel image?
How are the attention scores mapped into the heat map.
How do you perform Bag classifier, in Figure 7?
IN BRIEF. THE COMPUTATIONS HAPPENING IN FIGURE 3 AFTER THE
PATCH FEATURES NEED TO BE EXPLAINED IN FURTHER DETAIL AND COMPLEMENTED WITH THE INPUT-OUT SUBPROCESSES DESCRIPTIONS
AND THEIR PARAMETERS.
A fixed seed for the random number generator is not a good idea since
nothing becomes random afterwards. Every new run a new seed must be computed from the machine clock and set to generate new random numbers.
One last question I would like you to explain is the training set size.
Line 477: the training set size is 75% , the rest 25% is split in 2 equal size sets for validation and test. Wow, 75% is really a big chunk of the available data.
The use of this huge set seems to indicate that something was not working out
as expected. Accordingly with thousand NN authors, the training and validation sets take about 40% of the available data. Besides, your approach is presented as a technique to deal with the scarcity of the data. I expected to read the characteristics of such a system when dealing with the training set.
Reviewer 2 Report
The paper covers an interesting and relevant work in the field of breast cancer diagnosis. The introduction section provided sufficient information to navigate through the work and understand the proposed method and corresponding results. The study design is well thought out and provides a solid backbone for the work, which adds to the credibility of the findings. The results were presented in a clear and concise manner, and the use of appropriate visual elements helped to emphasize the key findings. Finally, the discussion and concluding remarks nicely wrapped up the work and provided a clear summary of the main findings and their implications. Overall, the paper is well-structured and clearly written.
Round 2
Reviewer 1 Report
I have no further comments.